# Perceived Importance of Types and Characteristics of Support to Informal Caregivers among Spouse Caregivers of Persons with Dementia in Sweden: A Cross-Sectional Questionnaire-Based Study

**DOI:** 10.3390/ijerph21101348

**Published:** 2024-10-11

**Authors:** Marcus F. Johansson, Kevin J. McKee, Lena Dahlberg, Christine L. Williams, Lena Marmstål Hammar

**Affiliations:** 1School of Health and Welfare, Dalarna University, 791 88 Falun, Sweden; kmc@du.se (K.J.M.); ldh@du.se (L.D.); 2Aging Research Center, Karolinska Institutet & Stockholm University, 171 65 Solna, Sweden; 3Christine E Lynn College of Nursing, Florida Atlantic University, Boca Raton, FL 33431, USA; cwill154@health.fau.edu; 4School of Health, Care and Social Welfare, Mälardalen University, 721 23 Västerås, Sweden; lena.marmstal.hammar@mdu.se

**Keywords:** spouse caregivers, dementia, community-living, caregiver support

## Abstract

Informal caregivers play a crucial role in the care of individuals with dementia, and their caregiving may significantly impact their own health and well-being. This cross-sectional survey study focuses on the perceived importance of various types and characteristics of formal support in a convenience sample of caregivers aged 65 years or older (N = 175) caring for a spouse with dementia. Participants completed a questionnaire containing 17 items describing different types of support and 12 items describing different characteristics of support, rating their importance. The questionnaire also contained questions on various caregiving-related factors. Principle components analysis (PCA) was carried out on the importance ratings, separately, on the types of support items and the characteristics of support items. Each PCA produced three components. For types of support, they were Proficiency and Opportunity, Supportive Structures, Flexible Counselling. For characteristics of support, they included Respectful and Competent, Timely Support, and Accessible and Acceptable. The three characteristics of the support components all had higher mean importance ratings than the three types of support components. The content of some components indicated that while spouse caregivers rate support for their caregiving needs as important, they may not always differentiate their own needs from those of their partner with dementia. The negative impact of caregiving was the factor most strongly and consistently associated with the components’ importance ratings. This study emphasizes the need for health and social care providers to address the unique needs of spouse caregivers while simultaneously ensuring the delivery of quality care for individuals with dementia.

## 1. Introduction

Globally, over 57.4 million persons are living with Alzheimer’s disease or other dementia disorders, of which 7.8 million live in Western Europe (GBD 2019 Dementia Forecasting Collaborators [1]). Most persons with dementia reside in the community, with the largest part of the support they receive provided by informal caregivers, such as adult children or spouses [2,3,4]. In Sweden, care of persons with dementia is most often provided by a spouse [5]. While spouse caregivers of persons with dementia may experience caregiving positively, they are also likely to experience negative impacts, such as psychological strain, declining health and financial burden [6,7,8], and are more vulnerable to such negative outcomes than other informal caregivers [9,10]. Support for the well-being of informal caregivers of persons with dementia needs to be strengthened [11], and the number of countries in Europe with policies on supporting dementia caregivers is growing [12]. Providing appropriate formal support to spouse caregivers of persons with dementia is imperative to ameliorate the negative impact of caregiving on the caregiver while also potentially enhancing the well-being of the person with dementia. This study explores how spouse caregivers of persons with dementia perceive the importance of different types and characteristics of formal support and the factors that are associated with their perceptions.

Formal support services or programs (e.g., counseling, education, respite care, day care) can be effective in meeting the needs of caregivers [13,14]. However, the evidence for their effectiveness in moderating negative caregiver outcomes is unclear [13,15] and the uptake of services such as counselling and support groups is generally low [10,16,17]. Factors positively associated with support use include the care-recipient being male and able to be left alone for a few hours, the caregiver being female, a stressful caregiving situation, living in areas with service providers in close proximity, and social support in the community [17,18,19]. Conversely, studies have identified a range of barriers to caregivers accessing or using support, such as time constraints and a lack of information about or trust in a service [18,20]. Characteristics of support, such as positive previous encounters with support providers, may promote use, while a lack of flexibility in support or the perception that the service is unnecessary or undesirable may hinder access or use of support [20,21,22]. Often, spouse caregivers can prioritize the needs of the person with dementia over their own support needs [23,24]. Research on caregivers’ experiences of support has tended to focus on specific services, and there is a need for more research that considers caregivers’ views on support more broadly [13,25].

In Sweden, the number of beds in residential care have steadily decreased since the 1990s, and the number of informal caregivers has increased in all social groups [26]. While social services in Sweden are obligated to support informal carers [27], only a minority of spouse caregivers receive support [10]. A better understanding of the types and characteristics of support spouse caregivers of persons with dementia find important is a necessary foundation both for developing services grounded in the caregiver’s own priorities and for increased uptake of support. Knowledge on how the perceived importance of types and characteristics of support is associated with factors related to the caregiver, the care-recipient, and the broader caregiving context, can also help social workers and care professionals to better target existing support and to develop new services for spouse caregivers in specific situations or with specific needs.

Thus, this study aims to explore the perceived importance of types and characteristics of formal support among spouse caregivers of people with dementia. A secondary aim is to examine how perceived importance of support is associated with factors related to the caregiver, the caregiving situation and the relationship between the spouse caregivers and the person with dementia.

## 2. Materials and Methods

### 2.1. Study Design and Participants

The study was a cross-sectional, questionnaire-based survey, with data collected from August 2019 to March 2020 using convenience sampling. The main eligibility criterion for the study was that a potential participant should be a spouse caregiver of a person with dementia, defined in the study as “a person aged 65 years or older and living with a spouse or partner with a dementia disorder to whom they provide care, help or support”. Care, help or support was further defined as “efforts a person makes on a regular basis such as personal care, supervision, household activities and maintenance, transportation, or contacts with services, this can include supporting the care-recipient’s personal economy, paying invoices, etc.”. Additional eligibility criteria were proficiency in Swedish and living in ordinary housing.

Of the 175 persons who completed the questionnaire, 12 did not meet the above criteria, providing an analytical sample of N = 163. Due to the nature of the study’s recruitment process (see Procedure, below), the number of eligible persons who received a questionnaire is unknown and, thus, a response rate cannot be estimated.

### 2.2. Material and Measures

A self-completion questionnaire was developed, focusing on the following topics: caregiver and care-recipient demographics; caregiving situation; caregiver-related factors; caregiver–care recipient relationship; and perceived importance of caregiver support.

Where available, instruments and questions translated to and/or validated in Swedish were used. Where no Swedish version was available, instruments underwent standard translation procedures, including back translation.

#### 2.2.1. Demographic Variables

Questions addressed the respondent’s and care-recipient’s gender (Male (0), Female (1)); year of birth; year when the couple’s relationship was established; type of accommodation (house/townhouse (1), apartment (0)); and municipality grouped (large cities and municipalities near large cities (3), medium-sized towns and municipalities near medium-sized towns (2); smaller towns/urban areas, and rural municipalities (1)) [28].

#### 2.2.2. Caregiving Situation

Questions addressed the care-recipient’s dementia diagnosis, years since receiving diagnosis and years as a caregiver.

Caregiving intensity was measured by an open-ended question: “In a normal week, how many hours do you provide care?”.

The Behavioral and Instrumental Stressors in Dementia (BISID) [29] instrument was used to measure the level of caregiver stress due to providing care to a partner with dementia. BISID measures (a) the caregiver-reported frequency of occurrence of a series of behavioral problems related to dementia and (b) the caregiver’s perceived level of stress due to each behavioral problem. In this paper we report the data on perceived stress only. BISID contains the question “Does the person you care for behave in any of the following ways…” followed by a list of 12 behavioral problems. The respondent rates the level of stress experienced due to each problem on a four-point scale, from Not Stressful (0) to Very Stressful (3). Sample Cronbach’s alpha for behavioral stress: α = 0.82.

The caregivers’ perception of their caregiving situation was measured with the COPE Index [30,31], using its subscales for Negative Impact (7 items) and Positive Value (4 items). Sample Cronbach’s alpha for the two subscales: Negative Impact, α = 0.82; Positive Value, α = 0.70.

#### 2.2.3. Caregiver-Related Factors

Self-rated health was assessed using a single item [32]: “How would you rate your general health?”, with a five-point response scale from Excellent (0) to Poor (4).

Loneliness was assessed using the 6-item de Jong Gierveld Loneliness Scale [33]. The scale is composed of two subscales: emotional loneliness (3 items) and social loneliness (3 items) with response options No (0), More or Less (1) and Yes (1). Sample Cronbach’s alpha for the two subscales: emotional loneliness, α = 0.57; social loneliness, α = 0.79.

The degree to which caregivers perceive that their life is full of meaning was assessed with the Presence of Meaning subscale from the Meaning in Life Questionnaire [34]. The subscale contains 5 items, with response options on a seven-point scale from Absolutely Untrue (1) to Absolutely True (7). Sample Cronbach’s alpha for the subscale: α = 0.87.

#### 2.2.4. Relationship Quality

The relationship between caregiver and care-recipient was assessed using the Mutuality scale [35]. The scale contains 15 items, with response options on a four-point scale ranging from Not Very Much (1) to A Great Deal (4). Sample Cronbach’s alpha: α = 0.94.

Changes in the relationship were assessed by two items [8]: “How emotionally close do you feel to your partner today, compared with before he/she developed dementia?” and “How satisfied are you with the physical intimacy with your partner today, compared with before he/she developed dementia?” Response options were: Less Close/Satisfied (−1), Unchanged (0), More Close/Satisfied (1).

#### 2.2.5. Perceived Importance of Support

Perceived importance of support was measured using items that originated in a study in the United Kingdom [36] that used a qualitative approach to explore informal caregivers’ perceptions of the formal support they received, with a particular focus on respite care. Findings from that study were adapted and used in the EUROFAMCARE project [37], where the adapted items measured caregivers’ perceived level of importance of formal support. Two aspects of support were addressed: different types and different characteristics. Perceived importance of types of support was measured with the question “How important is support that gives you …?” followed by a list of 17 types of support. Perceived importance of characteristics was measured with the question “How important are the following characteristics of services for you?” followed by 12 statements regarding service characteristics. Respondents were asked to indicate level of importance on a four-point response scale from Not Important (0) to Extremely Important (3) (see Appendix A).

### 2.3. Procedure

Study information describing the study’s aim, methods, and participant eligibility criteria was sent to organizations across Sweden through professional networks engaged in either dementia care or caregiver support. Organizations interested in helping to distribute the study questionnaire to potential participants were instructed to contact the research team. The 37 public health and social care providers and 2 civil society organizations that contacted the research team were subsequently sent questionnaire bundles. These contained informational flyers and a number of sealed envelopes each containing the study questionnaire, study information and a consent form. The organizations were asked to identify potential participants and provide each with a sealed envelope. The study information instructed potential participants to compete the questionnaire if they believed they met the provided eligibility criteria. They were further instructed to return the completed questionnaire and signed consent form by post to the research team. The questionnaire could optionally be completed via a postal survey (89%) in a web-based format (9.8%) or via a structured interview by telephone conducted by the first author (1.2%).

Information about the study was also made available on social media and media outlets, and potential participants who contacted the research team about the study were provided with a sealed envelope. Of the completed questionnaires received from eligible participants, 89.0% were postal, 9.8% web-based and 1.2% telephone interviews.

The study was conducted in accordance with the Declaration of Helsinki, and all participants were asked to provide informed consent upon participation. Further, in compliance with the Swedish Ethical Review Act (2003:460), this study received ethical approval by the Swedish Ethical Review authority 25 July 2019, (reg.no. 2019-03288).

### 2.4. Data Analysis

Descriptive statistics were performed on all variables to examine central tendency and dispersion and to describe the sample. Summative scores were calculated for the COPE index subscales and Presence of Meaning subscale, while mean scores were calculated for the BISID and Mutuality scales to retain cases. Subscales scores for Emotional and Social Loneliness were created following established guidelines.

Separately, items measuring perceived importance of types of support and items measuring perceived importance of characteristics of support were subjected to Principal Component Analysis (PCA). PCA is a method for reducing the dimensionality in a dataset, in which a large number of variables are condensed into a smaller number of components that are linear combinations of the original variables and retain much of the variation in the original data. Due to the importance of maintaining a high case-to-item ratio for PCA, pairwise deletion of cases with missing data was preferred to listwise deletion. The suitability of the data for PCA was assessed by the Kaiser–Mayer–Olkin measure of sampling adequacy and Bartlett’s test of sphericity, which were satisfactory for both type of support and characteristic of support items. The Kaiser criterion of an eigenvalue > 1.0 was used to determine component extraction, while Direct Oblimin rotation was applied to the extracted components, as the underlying constructs in the data could not be assumed to be independent. Following rotation, the components were assessed for satisfactory internal consistency reliability. The content of the components was then discussed among the research team and interpretative labels applied to them.

Bivariate associations between study variables and components of perceived importance were estimated using Kendall’s Tau-c, as several of the included study variables were on different scales or levels of measurement. For analyses of associations between components of perceived importance of support, Pearson’s product moment correlation coefficients were estimated.

Statistical significance for all analyses was set at *p* < 0.05. Due to multiple testing inflating the risk for cumulative Type I error, each significance test should be considered in relation to the obtained effect size.

## 3. Results

### 3.1. Descriptive Analyses

Table 1 presents descriptive statistics on spouse caregivers and care-recipients. Spouse caregivers were aged 65–89 years, and 76.7% were females. The care-recipients were aged 62–93 years, and a majority was male (78.5%). The duration of the caregiver–care recipient relationship ranged from 8 to 70 years (M = 48.61).

Almost half of spouse caregivers (49.1%) cared for a partner with Alzheimer’s disease dementia, followed by unspecified or mixed dementia (20.8%) and vascular dementia (19.5%). On average, the care-recipient had been diagnosed 3.20 years ago, and the caregiver had provided care for 4.40 years with a current mean of 71.51 h of care per week (Table 1).

On average, spouse caregivers reported no or low levels of stress due to their partners’ behavioral difficulties (M = 0.89, SD = 0.52, range 0–2.45). Responses to the COPE Index subscales indicate that spouse caregivers on average experienced both negative impacts (M = 15.31, SD = 3.90; range 7–28) and positive values (M = 11.47, SD = 2.48; range 4–16).

The average participant rated their health as fair, experienced both social and emotional loneliness and a presence of meaning and mutuality close to the scale-midpoint (M = 2.52, SD = 0.66, range 1.07–3.73) and reduced satisfaction with both emotional closeness and physical intimacy.

### 3.2. Perceived Importance of Support

The PCA of the items measuring types of support produced three components, explaining 58.8% of the variance: Proficiency and Opportunity (seven items), Supportive Structures (four items), and Flexible Counselling (five items). The PCA of the items measuring characteristics of support also produced three components, explaining 73.2% of the variance: Respectful and Competent (five items), Timely Support (three items) and Accessible and Acceptable (four items). Table 2 and Table 3 present the results of both PCAs, including the labelled components, descriptive statistics for items and rank order of item means within components, component item loadings, and Cronbach’s α for each component.

For the PCA of the items measuring perceived importance of types of support, item means ranged from 2.06 (SD = 0.76) for “Information about the dementia disease that partner has” to 0.70 (SD = 0.96) for “More money to help provide things I need to give good care”.

In the component Proficiency and Opportunity, four types of support were related to information or skill development needs of the caregiver, “Information about the dementia disease that partner has” had the largest component loading (0.85). The remaining three types of support were related to opportunities to undertake stimulating activities outside of caregiving for the spouse caregiver or care-recipient. The component Supportive Structures contained four types of support, with “Help to make partner’s environment more suitable for caregiving” having the largest component loading (0.77). Finally, Flexible Counselling contained five types of support relating to individual or group counselling with different types of delivery mode, with “The opportunity to talk online or on the phone over my problems as a caregiver with a professional” having the largest component loading (0.76) (see Table 2).

For the PCA of the items measuring perceived importance of characteristics of support, item means ranged from 2.52 (SD = 0.61) for “Care workers treat partner with dignity and respect” to 1.72 (SD = 0.81) for “The help provided fits in with your own routines”. The component Respectful and Competent contained five different characteristics of support, of which three were related to the need for either the spouse caregiver, the care-recipient or their situation to be dealt with respectfully, with two characteristics concerning care workers’ training needs for delivering high quality care. “Care workers treat partner with dignity and respect” had the largest component loading (0.91). The component Timely Support contained three characteristics relating to support availability and timeliness, with the item “The help provided fits in with your own routines” having the largest component loading (0.95). Finally, Accessible and Acceptable contained four characteristics of support relating to a focus on the spouse caregiver’s or care-recipient’s needs while also being accessible. The item “The help provided is not too expensive” had the largest component item loading (0.83).

#### 3.2.1. Bivariate Associations

Table 4 presents the bivariate associations between the components of perceived importance of support and demographic variables, caregiving situation, caregiver-related factors and relationship quality. Higher scores on components reflect greater perceived importance.

#### 3.2.2. Proficiency and Opportunity

Female gender of caregiver was associated with Proficiency and Opportunity, *τ_c_* (156) = 0.22, *p* = 0.008, while the component had a negative relationship with spouse caregivers’ age, *τ_c_* (155) = −0.13, *p* = 0.022. Higher caregiving intensity, *τ_c_* (121) = 24, *p* ≤ 0.001, behavioral stress, *τ_c_* (152) = 0.17, *p* = 0.003, negative impact due to caregiving, τc (153) = 27, *p* ≤ 0.001, and emotional loneliness, *τ_c_* (151) = 0.21, *p* = 0.020, were positively associated with Proficiency and Opportunity, while positive value of care, *τ_c_* (154) = −0.15, *p* = 0.011, and increased emotional closeness due to the care recipient’s dementia, *τ_c_* (154) = −0.16, *p* = 0.019, had negative associations with Proficiency and Opportunity.

#### 3.2.3. Supportive Structures

Female gender was significantly associated with Supportive Structures, *τ_c_* (150) = 0.19, *p* = 0.012. Caregiving intensity, *τ_c_* (116) = 0.18, *p* = 0.007, behavioral stress, *τ_c_* (147) = 0.19, *p* ≤ 0.001, and negative impact due to caregiving, *τ_c_* (148) = 0.27, *p* ≤ 0.001, were positively associated with Support Structures, while decreased intimacy due to the care recipients’ dementia was negatively associated with Supportive Structures, *τ_c_* (147) = −0.16, *p* = 0.010.

#### 3.2.4. Flexible Counselling

Caregiving intensity, *τ_c_* (120) = 0.19, *p* = 0.009, behavioral stress, *τ_c_* (150) = 0.21, *p* ≤ 0.001, negative impact due to caregiving, *τ_c_* (151) = 0.29, *p* ≤ 0.001, and emotional loneliness, *τ_c_* (150) = 0.30, *p* ≤ 0.001, were positively associated with Flexible Counselling. All measures of relationship quality (mutuality, increased closeness, and increased intimacy due to the care-recipient’s dementia) all had negative associations with Flexible Counselling (*τ_c_* = −0.20 to τc = −0.12).

#### 3.2.5. Respectful and Competent

Female gender had a positive association with the component Respectful and Competent, *τ_c_* (152) = 0.17, *p* = 0.042, while greater age had a negative association, *τ_c_* (151) = −0.17, *p* = 0.001. Years since dementia diagnosis, *τ_c_* (142) = 0.14, *p* = 0.016, caregiving intensity, *τ_c_* (118) = 0.22, *p* = 0.001, behavioral stress, *τ_c_* (148) = 0.13, *p* = 0.021, and negative impact due to caregiving, *τ_c_* (150) = 0.12, *p* = 0.040, were positively associated with Respectful and Competent.

#### 3.2.6. Timely Support

Female gender had a positive association with Timely Support, *τ_c_* (154) = 0.16, *p* = 0.031, while caregiver age had a negative association, *τ_c_* (153) = −0.17, *p* = 0.003. Caregiving intensity, *τ_c_* (118) = 0.27, *p* ≤ 0.001, and negative impact due to caregiving, *τ_c_* (152) = 0.15, *p* = 0.015, had positive associations with Timely Support.

#### 3.2.7. Accessible and Acceptable

Female gender was associated with the component Accessible and Acceptable, *τ_c_* (153) = 0.24, *p* = 0.003, as were years since dementia diagnosis, *τ_c_* (142) = 0.18, *p* = 0.004, caregiving intensity, *τ_c_* (118) = 0.18, *p* = 0.005, behavioral stress, *τ_c_* (149) = 0.19, *p* ≤ 0.001, negative impact due to caregiving, *τ_c_* (151) = 0.22, *p* ≤ 0.001, self-reported health, *τ_c_* (152) = 0.16, *p* = 0.012, and emotional loneliness, *τ_c_* (149) = 0.30, *p* ≤ 0.001; presence of meaning, *τ_c_* (147) = −0.12, *p* = 0.036 and mutuality, *τ_c_* (150) = −0.12, *p* = 0.029, had negative associations with the component.

## 4. Discussion

This study explored how spouse caregivers of persons with dementia perceive the importance of different types and characteristics of formal support, and examined how these perceptions are associated with factors related to the caregiver, the care-recipient and the caregiving situation. We found three components of perceived importance of types of support: Proficiency and Opportunity, Supportive Structures, and Flexible Counselling; and three components of perceived importance of characteristics of support: Respectful and Competent, Timely Support, and Accessible and Acceptable. Many of the factors measured in our study were associated with some or most of the components, although most associations were small-to-moderate in strength. Factors that might be regarded as indicators of caregiving burden were commonly positively associated with perceived importance on all, or nearly all, of the components, e.g., caregiving intensity, behavioral stress, and negative impact of caregiving. Factors indicative of a positive experience of care, such as mutuality, positive value, and increased emotional closeness, were commonly negatively associated with perceived importance on one or more components. Gender was associated with five of the six components, with female compared to male caregivers having higher levels of perceived importance; both emotional loneliness and caregiver age were associated with higher perceived importance on three. The Accessible and Acceptable component was associated with nine factors, while Proficiency and Opportunity was associated with eight; Timely support was associated with only four factors, the lowest number of associations of all the components. Below, we elaborate on our findings in more detail.

How informal caregivers perceive the importance of formal support services has rarely been studied, with formal support usually conceptualized from a care-professional or organizational perspective in terms of the mode of delivery or targeted outcome for support [38,39]. From our findings, this way of conceptualizing support seems simplistic. Several of our components described support targeted to both the spouse caregiver and care-recipient. The component Proficiency and Opportunity provides a clear example where seemingly unrelated items concerning caregiver training and education align with meaningful activities for the care-recipient and respite. These findings highlight how spouse caregivers’ perceptions of support may differ from those of professionals and suggest that developing support that integrates content relating to both the caregiver and care-recipient may be of value. The items loading on the Proficiency and Opportunity component would suggest that spouse caregivers see both caregiver education and a well looked-after care-recipient as important and related, and support that seeks to combine these objectives would arguably be of considerable value and facilitate service use.

Considering the components’ mean scores for importance, Respectful and Competent had the highest mean (2.31) while Supportive Structures had the lowest (0.97). Relating these means back to the scale on which responses to the individual questions were measured, all components would appear to be perceived as important, but some considerably more important than others. Interestingly, the means for all three components derived from the questions on types of support were lower than those for the three components derived from questions on characteristics of support. One interpretation of this finding is that the content and quality of support is of greater concern to caregivers than its form and purpose. As an example, the item “care workers treat the person with dementia with dignity and respect”, from the component Respectful and Competent, had the highest mean score of all items for perceived importance. This item does not specify a particular support service or the purpose of the support, but rather focuses on how the support should be experienced. The high importance rating of this item also indicates how spouse caregivers often give the needs of the care-recipient more importance than their own needs. Spouse caregivers’ lack of self-focus has been discussed in previous research [23], with our results suggesting that spouse caregivers’ perceptions of support to them as caregivers is for the most part entwined with their perceptions of support directed to the care-receiving partner. The need to focus on the spouse caregiver as separate from the couple unit, as well as the need for more couple-centered approaches, have been considered and critically discussed in relation to support and in dementia care [40,41,42,43]. A couple-centered approach includes working both with the individual needs of the spouse caregiver and those of the person with dementia, as well as their shared needs as a couple.

As might be anticipated, those factors indicative of a demanding caregiving situation were consistently associated with higher perceived importance of support, while those factors indicating a positive care experience and a good relationship between caregiver and care-recipient were often associated with lower perceived importance. A demanding caregiving situation has been shown to hinder support use [20,22]. Spouse caregivers may therefore perceive support as important while still being restricted from accessing support due to their circumstances. Providing the caregiver with an opportunity to leave the care-recipient safely for a few hours has been shown to facilitate formal support use [19,20], while promoting and supporting the relationship between the spouse caregiver and the care-recipient can promote resilience and lead to the caregiving situation being experienced as less burdensome [43,44,45]. In our study, reductions in physical intimacy and emotional closeness in the caregiver–care recipient relationship were both negatively associated with two components of perceived importance, suggesting there can be value in support that helps spouse caregivers manage conflicting and contradictory feelings towards the care-recipient that can occur as dementia progresses [42,46]. Such findings underline the relevance of couple-centered approaches to dementia care.

Previous research has found that spouse caregivers tend to experience loneliness more than other caregivers and other older adults [47,48]. In our study, emotional loneliness was positively associated with three components of perceived importance of support, but social loneliness had no significant association with any component. This is in line with socioemotional selectivity theory, which proposes that with ageing, the satisfaction of emotional needs becomes more important than meeting social needs [49,50]. Emotional loneliness was most strongly associated with the components Flexible Counselling and Accessible and Acceptable, components that include support or opportunities for caregivers to meet others in similar circumstances or care professionals, to talk about and discuss their situation and feelings. Our findings reinforce previous research suggesting peer-support and emotional and educational interventions as possible ways to target emotional loneliness among spouse caregivers of persons with dementia [51]. Female compared to male spouse caregivers had higher scores for perceived importance on five out of six components, suggesting that female spouse caregivers see value in support to a greater extent than males. Male spousal caregivers have been shown to be more reluctant to seek support than female caregivers and may have different preferences [52], while the effectiveness of different caregiver support interventions can differ between male and female caregivers [53]. Such findings may well reflect the influence of social norms and stereotypes of females as “caregivers” [54] on how support is accessed, used and responded to, including its perceived importance [55]. Caregiver age was also negatively associated with three components of perceived support, suggesting that young–old caregivers may perceive more value in support than old–old caregivers. Since the duration of being a caregiver was not associated with any component, this finding may reflect a cohort effect or a true ageing effect. More research is required into how support can be better tailored to gender- and age-related needs and preferences.

### Study Strengths and Limitations

Key strengths of this study are its originality and sharp focus. Few studies have explored how caregivers perceive the importance of types and characteristics of support, with most research based on organizational or professional classifications of support. From the ratings of importance provided by the caregivers in our study, we were able to produce a conceptual map of support from the caregiver’s perspective. Importantly, our focus was on perceptions of support rather than on support use, as types and characteristics of support may be perceived important even if they have not been accessed previously or are not available. We restricted our study to spouse caregivers of people with dementia, and while that means we cannot generalize to the wider population of informal caregivers, we can be more certain of the applicability of our findings for this important group of caregivers. We also explored a wide range of factors to understand how they might be related to the perceived importance of support, relating to the caregiver and care-recipient, the caregiving situation, and the caregiver–care recipient relationship. However, with the study’s focus primarily on the caregiver, there was a relative lack of questions that explored the needs of the care-recipient and how the perceived importance of support might vary with those needs. Future research could build on the present study by collecting more comprehensive data that encompass the needs and perceived importance of support of both caregiver and care-recipient.

There are limitations to the study. First, the use of a cross-sectional design means that we cannot assume that the observed associations in the data are casual. Secondly, the relatively small convenience sample of spouse caregivers implies that any generalization of the study findings should be handled with caution. For example, the sample may not include spouse caregivers prevented from participating as a result of the demands of their caregiving situation. In addition, the study eligibility criterion of proficiency in Swedish will have excluded some caregivers from a migrant or ethnic minority background with a poor command of Swedish, which further limits the potential for generalization of our findings. The sample was, however, large enough to provide a sufficient case-to-item ratio for good factorability for PCA, which produced an interpretable component structure. Yet, the sample was too small to allow for confirmatory factor analysis or subgroup analysis.

Cronbach’s alpha was relatively low for some scales, particularly the Emotional Loneliness subscale. Cronbach’s alpha for a scale tends to increase with the number of items the scale contains. Since the Emotional Loneliness subscale consists of only three items, one might anticipate an alpha for the subscale lower than optimal, but an alpha of 0.57 is disappointing. The mean of the interitem correlations for the scale was estimated and found to be satisfactory, but the low alpha might still suggest that the associations between the Emotional Loneliness subscale and other variables are not equally influenced by all scale items.

Further, while bivariate analysis revealed several significant associations between study variables and components, these associations should be interpreted with caution as the potential effect of other variables on the associations has not been controlled. In addition, and as mentioned in the Materials and Methods section, the number of bivariate analyses means that there is an increase in the risk of Type I error due to multiple testing, and as such, each association needs to be considered in relation to the obtained effect size.

Finally, some variables had a relatively high number of missing cases. Variation in internal missing cases results in a reduced sample size for specific analyses, which in turn increases the potential for Type II error. In addition, variation in non-response may reflect the presence of unknown factors that have the potential to bias our study findings.

## 5. Conclusions

While spouse caregivers have their own needs of support, they do not necessarily experience support to themselves as more important than support directed to their partner with dementia; thus, researchers and practitioners need to have a holistic approach towards spouse caregivers, focused on the spouse caregiver’s needs while acknowledging the care needs of the partner to support the spouse caregiver.

Furthermore, the demands of the caregiving situation are a key factor in understanding what may influence spouse caregivers perceptions of support. Yet, there is variability among spouse caregivers and how characteristics such as gender and age may influence their perceptions of support.

## Figures and Tables

**Table 1 ijerph-21-01348-t001:** Sample characteristics and background variables on spouse caregivers and care-recipient partners (N = 163).

	Mean, (SD), Range	%
Demographic variables		
Age, spouse carer (n = 161)	75.30 (5.82), 65–89	
Gender, spouse carer (n = 163)		
Female		76.7
Age, care-recipient (n = 160)	78.22 (6.67), 62–93	
Gender, care-recipient (n = 163)		
Female		21.5
Years in relationship (n = 160)	48.61 (13.44), 8–70	
Years of co-residency (n = 155)	46.30 (13–23), 7–69	
Type of accommodation (n = 151)		
House/townhouse		54.3
Classification of municipality of residence		
Large cities and municipalities near large cities		19.0
Medium-sized towns and municipalities near medium-sized towns		42.3
Smaller towns/urban areas and rural municipalities		38.7
Caregiving situation		
Years since dementia diagnosis (n = 152)	3.20 (2.92), 1–20	
Received dementia diagnosis (n = 159)		
Alzheimer’s disease		49.1
Vascular dementia		19.5
Dementia with Lewy bodies		5.7
Frontotemporal dementia		3.1
Dementia due to Parkinson’s disease		1.9
Unspecified or mixed dementia		20.8
Years as a carer (n = 156)	4.40 (4.52), 1–43	
Hours of care/average week (n = 124)	71.51 (61.64), 1–168	
Behavioral stress (n = 156)	0.89 (0.52), 0–2.45	
Negative impact (n = 159)	15.31 (3.90), 7–28	
Positive value (n = 161)	11.47 (2.48), 4–16	
Caregiver-related factors		
Self-rated health (n = 160)	2.39 (0.95), 0–4	
Emotional loneliness (n = 157)	1.57 (0.97), 0–3	
Social loneliness (n = 158)	1.68 (1.24), 0–3	
Presence of meaning (n = 155)	23.89 (6.58), 5–35	
Relationship quality		
Mutuality (n = 158)	2.52 (0.66), 1.07–3.73	
Change in emotional closeness (n = 159)	−0.48 (0.69), −1–1	
Change in physical intimacy (n = 159)	−0.75 (0.46), −1–1	

**Table 2 ijerph-21-01348-t002:** Descriptive statistics and principal components for perceived importance of types of support; sample level rank of means. Scale n, alpha, mean, standard deviation and component loadings (N = 163).

	Rank	*M*	*SD*	Range	*n*	*α*	Component Item Loading
Component 1: Proficiency and Opportunity		1.84	0.67	0.29–3	158	0.86	
Information about the dementia disease that partner has	1	2.06	0.76	0–3			0.85
Information and advice about the type of help and support that is available and how to access it	2	1.96	0.77	0–3			0.77
Training to help me develop the skills I need to care	8	1.71	0.92	0–3			0.71
Opportunities to enjoy activities outside of caring	5	1.84	0.99	0–3			0.48
Help with planning for the future care	9	1.63	0.88	0–3			0.43
Opportunities for partner to undertake activities he/she enjoys	6	1.80	0.93	0–3			0.43
Opportunities to have a holiday or take a break from caregiving	3	1.92	1.02	0–3			0.42
Component 2: Supportive Structures		0.97	0.72	0–3	152	0.71	
Help to make partner’s environment more suitable for caregiving	13	1.18	1.00	0–3			0.77
Help to deal with family disagreements	16	0.72	0.95	0–3			0.74
Opportunities to spend more time with my family	12	1.33	1.03	0–3			0.68
More money to help provide things I need to give good care	17	0.70	0.96	0–3			0.57
Component 3: Flexible Counselling		1.49	0.78	0–3	156	0.84	
The opportunity to talk online or phone over my problems with a professional as a caregiver	11	1.38	1.02	0–3			0.76
Opportunities to attend a caregiver support group online or phone led by a professional	14	0.93	1.00	0–3			0.75
Opportunities to attend a caregiver support group meeting place for couples in the same situation	10	1.47	1.03	0–3			0.67
The opportunity to talk over my problems with a professional as a caregiver	7	1.80	0.96	0–3			0.66
Opportunities to attend a caregiver support group close to home lead by a professional	4	1.86	0.95	0–3			0.58

**Table 3 ijerph-21-01348-t003:** Descriptive statistics and principal components for perceived importance of characteristics of support; sample level rank of means. Scale n, alpha, mean, standard deviation and component loadings (N = 163).

	Rank	*M*	*SD*	Range	*n*	*α*	Component Item Loading
Component 1: Respectful and Competent		2.37	0.57	0.20–3.0	154	0.91	
Care workers treat partner with dignity and respect	1	2.52	0.61	0–3			0.91
Your views and opinions are listened to	2	2.39	0.62	1–3			0.90
The help provided improves the quality of life of partner	3	2.36	0.64	0–3			0.80
Care workers treat you with dignity and respect	4	2.33	0.69	0–3			0.75
Care workers have the skills and training they require	6	2.25	0.78	0–3			0.54
Component 2: Timely Support		1.93	0.65	0.33–3.0	156	0.78	
The help provided fits in with your own routines	12	1.72	0.81	0–3			0.95
Help arrives at the time it is promised	10	2.03	0.79	0–3			0.69
Help is available at the time you need it most	9	2.04	0.73	0–3			0.61
Component 3: Accessible and Acceptable		2.14	0.67	0.25–3.0	155	0.83	
The help provided is not too expensive	11	1.90	0.97	0–3			0.83
Help focuses on your needs as well as those of partner	8	2.14	0.80	0–3			0.79
The help provided improves your quality of life	5	2.28	0.70	0–3			0.58
Help is provided by the same care worker each time	7	2.22	0.79	0–3			0.55

**Table 4 ijerph-21-01348-t004:** Bivariate associations between perceived importance of types of support and characteristics of support and selected study variables.

	Index Proficiency and Opportunity		Index Supportive Structures		Index Flexible Counselling		Index Respectful and Competent		Index Timely Support		Index Accessible and Acceptable	
Variable	*τ_c_*	*p*	*τ_c_*	*p*	*τ_c_*	*p*	*τ_c_*	*p*	*τ_c_*	*p*	*τ_c_*	*p*
Female gender Spouse carer ^a^	0.22	0.008	0.19	0.012	0.14	0.074	0.17	0.042	0.16	0.031	0.24	0.003
Age spouse carer	−0.13	0.022	−0.08	0.143	−0.11	0.066	−0.17	0.001	−0.17	0.003	−0.06	0.310
Accommodation	0.17	0.065	0.04	0.713	−0.00	0.981	0.11	0.264	0.02	0.829	0.03	0.781
Municipality	0.13	0.057	0.10	0.183	0.03	0.721	−0.04	0.577	0.13	0.067	−0.10	0.184
Years since dementia diagnosis	0.08	0.191	0.07	0.263	0.08	0.217	0.14	0.016	0.12	0.066	0.18	0.004
Years as carer	0.09	0.153	0.05	0.393	0.10	0.102	0.03	0.614	0.05	0.419	0.10	0.114
Caregiving intensity	0.24	<0.001	0.18	0.007	0.19	0.009	0.22	0.001	0.27	<0.001	0.18	0.005
Behavioral stress ^b^	0.17	0.003	0.19	<0.001	0.21	<0.001	0.13	0.021	0.11	0.085	0.19	<0.001
Negative impact ^c^	0.27	<0.001	0.27	<0.001	0.29	<0.001	0.12	0.040	0.15	0.015	0.22	<0.001
Positive value ^c^	−0.15	0.011	−0.08	0.172	−0.10	0.079	−0.05	0.369	−0.09	0.142	−0.11	0.079
Self-reported health	0.03	0.626	0.04	0.488	0.00	0.971	0.04	0.515	0.05	0.431	0.16	0.012
Emotional loneliness ^d^	0.21	0.020	0.14	0.133	0.30	<0.001	0.06	0.497	0.10	0.267	0.30	<0.001
Social loneliness ^d^	0.07	0.472	−0.03	0.724	0.03	0.754	0.09	0.304	0.06	0.533	0.16	0.085
Presence of meaning ^e^	−0.01	0.923	−0.01	0.907	−0.04	0.491	−0.03	0.637	0.02	0.730	−0.12	0.036
Mutuality ^f^	−0.10	0.078	−0.07	0.213	−0.12	0.026	−0.01	0.825	−0.05	0.429	−0.12	0.029
Change in emotional closeness ^g^	−0.16	0.019	−0.07	0.310	−0.15	0.033	−0.04	0.617	−0.07	0.317	−0.07	0.322
Change in intimacy ^g^	−0.08	0.193	−0.16	0.010	−0.20	<0.001	−0.08	0.174	−0.05	0.367	−0.06	0.344

Note: ^a^ “male” = 0, “female” = 1; ^b^ BISID Behavioral stress section; ^c^ COPE Index; ^d^ de Jong Gierveld Loneliness scale; ^e^ Meaning in Life Questionnaire, Presence of meaning subscale; ^f^ Mutuality scale; ^g^ “less” = −1, “unchanged” = 0, “more” = 1; for analyses, n varies between 118 and 158 due to internal missing values.

## Data Availability

The data supporting the findings of this study are available on request from the corresponding author, MFJ. The data are not publicly available due to restrictions in accordance with the Swedish Public Access to Information and Secrecy Act (2009:400) and the EU General Data Protection Regulation (GDPR, 2016/679).

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
