# Peer review of "Perceived Importance of Types and Characteristics of Support to Informal Caregivers among Spouse Caregivers of Persons with Dementia in Sweden: A Cross-Sectional Questionnaire-Based Study"

_ijerph, 2024, doi:10.3390/ijerph21101348_

Round 1
Reviewer 1 Report
Comments and Suggestions for Authors
This is an interesting study examining the perceived importance of types and characteristics of support as perceived by spousal caregivers of persons with dementia in Sweden.
An important finding of this study is that the support should target both the caregivers and their care recipients, as well as the environment surrounding them. However, there are only 5 questions items (out of 29 items) which directly relate to care recipients. These include Opportunities for partner to undertake activities he/she enjoy, Help to make partner’s environment more suitable for caregiving; Care workers treat partner with dignity and respect; The help provided improves the quality of life of partner, as well as Help focuses on your needs as well as those of partner. A more comprehensive framework of questions which examine support on different needs of the care recipients in addition to those for caregivers would provide a more wholistic understanding of this argument. The current lack of such framework could be taken as a limitation of this study, or it can lead to a recommendation for future studies to examine more comprehensively needs and perceived importance of both caregivers and care recipients.
The manuscript has analyzed the items with different degree of loading under each component, and very often identified the one with the highest loading. This is appropriate. On the other hand, it would also be important to compare the degree of importance of the 6 components as perceived by the respondents, e.g., by analyzing the mean of each component, and to interpret the implications of these findings on support measures to caregivers. It is also interesting to find that the mean of supportive components index was relatively low, yet this component was seen as important. Are there any interpretation on this observation?
There are other areas which require relatively minor improvement/ amendments.
· p.2 ‘uptake of such services is low’: to provide useful background information, I would suggest the authors to list the types of services, and the uptake statistics of different services.
· P.4 Cronbach’s alpha for Emotional loneliness α = .57 was rather low. What might be the reasons? Emotional closeness was found to be more significant than social closeness in your study, so the low value deserves a bit more discussion on how this may affect the interpretation of the findings.
· P.4 How was the telephone interview being conducted? Was there an interviewer? If so, any measures to reduce interviewer bias. This is important because other forms of data collection (by post or by website) of this study did not involve any interviewer.
Author Response
Thank you for your comments on our manuscript. You'll find our responses bellow. All changes in the manuscript have been made with track changes.
Comment 1: An important finding of this study is that the support should target both the caregivers and their care recipients, as well as the environment surrounding them. However, there are only 5 questions items (out of 29 items) which directly relate to care recipients. These include Opportunities for partner to undertake activities he/she enjoy, Help to make partner’s environment more suitable for caregiving; Care workers treat partner with dignity and respect; The help provided improves the quality of life of partner, as well as Help focuses on your needs as well as those of partner. A more comprehensive framework of questions which examine support on different needs of the care recipients in addition to those for caregivers would provide a more wholistic understanding of this argument. The current lack of such framework could be taken as a limitation of this study, or it can lead to a recommendation for future studies to examine more comprehensively needs and perceived importance of both caregivers and care recipients.
Response 1: This is a valuable comment, and we have added material on this point to our strengths and limitations section in our Discussion, p. 15
Comment 2: The manuscript has analyzed the items with different degree of loading under each component, and very often identified the one with the highest loading. This is appropriate. On the other hand, it would also be important to compare the degree of importance of the 6 components as perceived by the respondents, e.g., by analyzing the mean of each component, and to interpret the implications of these findings on support measures to caregivers. It is also interesting to find that the mean of supportive components index was relatively low, yet this component was seen as important. Are there any interpretation on this observation?
Response 2: These are good observations, and it is clearly an omission that we have not reflected on the mean importance ratings of the components in the Discussion. We have added some material on this topic, including a reflection on what the variation in mean importance ratings of the components might indicate, p. 13. This finding has now also been included in the abstract.
Comment 3: p.2 ‘uptake of such services is low’: to provide useful background information, I would suggest the authors to list the types of services, and the uptake statistics of different services.
Response 3: The sources cited provide some of the detail the reviewer requests, which we feel if incorporated here would make the Introduction overly lengthy. However, ewe have added two examples and provide a further reference to research that describes receipt/use of support. P.2
Comment 4: P.4 Cronbach’s alpha for Emotional loneliness α = .57 was rather low. What might be the reasons? Emotional closeness was found to be more significant than social closeness in your study, so the low value deserves a bit more discussion on how this may affect the interpretation of the findings.
Response 4: Given that the subscales of the 6-item de Jong Gierveld Loneliness Scale both contain only three items, and that alpha usually increases with the number of items in a scale, one might anticipate that alpha for these subscales might be lower than optimal, but we agree the alpha for the Emotional Loneliness subscale is particularly disappointing. We have added a comment on this in the strengths and limitations section of the Discussion, p. 15. However, we are not sure how you arrive at your interpretation that ‘emotional closeness was found to be more significant than social closeness’, or whether you are referring to emotional/social loneliness or to our measures of change in emotional closeness and change in physical intimacy. Due to our uncertainty, we have not revised the manuscript in relation to this aspect of your comment.
Comment 5: P.4 How was the telephone interview being conducted? Was there an interviewer? If so, any measures to reduce interviewer bias. This is important because other forms of data collection (by post or by website) of this study did not involve any interviewer.
Response 5: On p 5 we now clarify that all telephone interviews were conducted by the first author, so there was no potential for inter-interviewer bias. As the interview was structured (i.e., followed the questionnaire exactly) and only 1.2% of questionnaires were complete this way, we feel the potential for biased data is low.
Reviewer 2 Report
Comments and Suggestions for Authors
I reviewed this manuscript with great interest, as it addresses a crucial topic concerning the types and characteristics of support for spouses who care for individuals with dementia. Please address the following comments:
-
Abstract It is essential that the abstract provides a clear and comprehensive overview of the study. Specifically, ensure that the background, methods, results, and conclusions are clearly articulated. The current abstract lacks sufficient detail in both the methods and results sections. The results should present the actual key findings rather than merely interpretative statements.
-
Introduction
- Consider consolidating the second and third paragraphs to enhance coherence.
- The fourth paragraph should be revised for greater conciseness.
-
Materials and Methods
- Line 94: The final number of participants should be reported in the Results section.
-
Results
- The principal component analysis seems to reveal multiple components. Could you explain why the sample sizes vary across different components? The reviewer presumes that the principal component analysis was conducted using a complete dataset, excluding cases with missing data. Is this assumption incorrect?
- The main analysis shows a varying sample size between 118 and 158, which could be influenced by approximately 20% of the participants, potentially introducing bias.
- Is correlation analysis adequate for your study? Please provide a rationale for not employing multivariate analysis techniques, such as multiple regression analysis.
Author Response
Thank you for your comments, all revisions have been made with track changes.
Comment 1: Abstract It is essential that the abstract provides a clear and comprehensive overview of the study. Specifically, ensure that the background, methods, results, and conclusions are clearly articulated. The current abstract lacks sufficient detail in both the methods and results sections. The results should present the actual key findings rather than merely interpretative statements.
Response 1: We have revised the abstract to include more information on the study Method and Results.
Comment 2: Introduction
Consider consolidating the second and third paragraphs to enhance coherence.
The fourth paragraph should be revised for greater conciseness.
Response 2: We have moved one sentence from the second paragraph to the first paragraph,
and shortened the third paragraph with focus on presenting what is essential information on the context of our study. Comment 3: Materials and Methods
Line 94: The final number of participants should be reported in the Results section.
The number of cases for each analysis is reported in the Results section, either in the heading of the table, in the body of the table, or in a note below the table.
Comment 4: Results
The principal component analysis seems to reveal multiple components. Could you explain why the sample sizes vary across different components? The reviewer presumes that the principal component analysis was conducted using a complete dataset, excluding cases with missing data. Is this assumption incorrect?
Response 4: Due to the importance of maintaining a high case-to-item ratio for PCA, pairwise deletion of cases with missing data was preferred to listwise deletion. We have now added a sentence clarifying this in the Method section, p. 5.
Comment 5: The main analysis shows a varying sample size between 118 and 158, which could be influenced by approximately 20% of the participants, potentially introducing bias.
In the bivariate analyses, the sample size for each analysis largely varied between 142 and 158. However, for the variable measuring caregiving intensity, ‘in an average week how many hours do you provide care?’ the sample sizes for the analyses varied between 118 och 123.We have added a comment on the issue in the strengths and limitations section of the Discussion. P.15
Comment 6: Is correlation analysis adequate for your study? Please provide a rationale for not employing multivariate analysis techniques, such as multiple regression analysis.
Response 6: Thank you for this comment. We did consider including multivariate analyses in the paper; but we felt the focus of the paper should be on the PCA and the content of the components, and that including regression models for each component both increased the length of the paper so that it became very data-dense while also altering the paper’s focus. Furthermore, trial runs of regression models of the components indicated that most of the variance in each model was explained by negative impact, while more nuance was provided by the bivariate analyses with regard to how the components vary in their associations with the measured factors.
Round 2
Reviewer 2 Report
Comments and Suggestions for Authors
Thank you for your diligent revisions to the manuscript. I appreciate the thoroughness of your responses to the comments. I have one observation to share: Comment 6: While I respect the authors' viewpoint, I wish to highlight certain concerns regarding bivariate analysis. Although bivariate analysis is straightforward in uncovering simple relationships, it can also facilitate misinterpretations driven by confounding factors. Moreover, the issue of multiple testing is critically significant when frequently employing bivariate analysis, as done in this study. My previous comment suggested the use of multivariate analysis to mitigate the risks associated with multiple testing. Therefore, it is imperative to explicitly address the limitations of bivariate analysis in the limitations section of your research.
Author Response
Comment 1: Thank you for your diligent revisions to the manuscript. I appreciate the thoroughness of your responses to the comments. I have one observation to share: Comment 6: While I respect the authors' viewpoint, I wish to highlight certain concerns regarding bivariate analysis. Although bivariate analysis is straightforward in uncovering simple relationships, it can also facilitate misinterpretations driven by confounding factors. Moreover, the issue of multiple testing is critically significant when frequently employing bivariate analysis, as done in this study. My previous comment suggested the use of multivariate analysis to mitigate the risks associated with multiple testing. Therefore, it is imperative to explicitly address the limitations of bivariate analysis in the limitations section of your research.
Response 1: We very much agree with the reviewer on this issue, and did mention the risk of multiple testing in the Method section (p. 6) of our manuscript. We have now also specified this issue in the study strengths and limitations subsection of the Discussion (p.15). In addition, we have added a sentence there on the potential for misinterpreting bivariate associations in the absence of control for the effect of other variables.